# Outcomes of Universal Newborn Screening Programs: Systematic Review

**DOI:** 10.3390/jcm10132784

**Published:** 2021-06-24

**Authors:** Christine Yoshinaga-Itano, Vinaya Manchaiah, Cynthia Hunnicutt

**Affiliations:** 1Institute of Cognitive Science, University of Colorado Boulder, UCB 594, Boulder, CO 80309, USA; Cynthia.Hunnicutt@colorado.edu; 2Department of Speech and Hearing Sciences, Lamar University, Beaumont, TX 77710, USA; vmanchaiah@lamar.edu

**Keywords:** childhood hearing loss, permanent childhood hearing loss, newborn hearing screening, universal hearing screening, early identification, early intervention, intervention outcomes

## Abstract

Background: This systematic review examined the outcomes (age of identification and intervention, developmental outcomes, cost-effectiveness, and adverse effects on parents) of universal newborn hearing screening (UNHS) for children with permanent congenital hearing loss (PCHL). Materials and methods: Multiple electronic databases were interrogated in March and April 2020 with further reports identified from article citations and unpublished literature. UNHS reports in English with comparisons of outcomes of infants who were not screened, and infants identified through other hearing screening programs. Results: 30 eligible reports from 14 populations with 7,325,138 infants screened through UNHS from 1616 non-duplicate references were included. UNHS results in a lower age of identification, amplification, and the initiation of early intervention services and better language/literacy development. Better speech perception/production were shown in younger, but not in older, children with early identification after UNHS. No significant findings were found for behavior problems and quality of life. UNHS was found to be cost-effective in terms of savings to society. In addition, no significant parental harm was noted as a result of UNHS. Conclusions: In highly developed countries, significantly better outcomes were found for children identified early through UNHS programs. Early language development predicts later literacy and language development.

## 1. Introduction

Studies from high-income countries (HICs) estimate that 1 in 1000 children are born with permanent childhood hearing loss (PCHL). However, children who are born in low/middle income countries (LMIC), as well as those children who were admitted to neonatal intensive care unit (NICU), have a much higher prevalence [1,2]. Despite the low prevalence, the PCHL could have severe negative consequences both at familial as well as societal levels.

Prior to establishing Universal Newborn Hearing Screening (UNHS) programs, the average language, literacy, social-emotional, and speech development of children with permanent childhood hearing loss (PCHL) was significantly lower than their peers with normal hearing. Eighteen-year-old children with hearing loss in the United States who were in the 12th grade between 1974–2003, had average literacy proficiency, between 3rd and 4th grade levels, more than two standard deviations below the developmental functioning of their hearing peers [3]. Wouters et al. [4] reported that deaf children in the Netherlands had the mean reading levels of first grade hearing students [2].

UNHS programs began to be implemented in the early 1990s, and by the end of the 1990s, there was evidence that these programs resulted in an earlier identification of hearing loss, earlier amplification, and earlier enrollment into early intervention services and that they significantly improved developmental outcomes in early childhood [5]. By 2007–8, evidence was deemed sufficient to established UNHS programs throughout most HICs and in some countries with fewer resources [6,7,8,9]. The quality benchmarks followed by most countries are those recommended by the Joint Committee on Infant Hearing [10,11]. However, due to a lack of endorsement from an international association such as the World Health Organization (WHO), countries make their own policy decisions. 

The Guidelines Development Group (GDG) of the WHO is responsible for developing clinical guidelines based on evidence which is then endorsed by the WHO. Following this, the WHO member countries are encouraged to adopt the recommendations in their own country and/or context. The GDG has recently been considering endorsing UNHS provided that there are positive outcomes reported by the academic literature examined using a systematic review. Due to a lack of any recent reviews on wider outcomes for UNHS, this review was commissioned. 

This systematic review was aimed at examining the outcomes of UNHS for children with PCHL. The key outcome indicators included: (a) early identification and intervention (i.e., age at dentification, age at amplification, age at intervention start), (b) developmental outcomes including receptive/expressive language, speech perception/production, literacy, social development, behavioral problems, and quality of life, (c) cost-effectiveness, and (d) adverse effects on parents of children with PCHL. 

## 2. Materials and Methods

This systematic review was prospectively registered with the International Prospective Register of Systematic Reviews (PROSPERO 2020 CRD42020175451) [12]. The review methods were guided by the Preferred Reporting Items for Systematic reviews and Meta-analyses (PRISMA) guidelines [13]. The authors of the study wrote the review protocol and performed all aspects of the review, although the WHO provided some input to the review design in order to ensure that the review met the key questions needed to be answered for the GDG to make a policy decision. 

### 2.1. PCHL Definition

All children with a bilateral hearing loss of 20 dB or greater in their better ear were included in the studies reviewed within this systematic review. 

### 2.2. Search Strategy

To identify eligible studies in March/April 2020, two reviewers (CYI, CH) interrogated electronic databases (PubMed, Medline (OVIDSP), Cochrane library, Google Scholar, Web of Science and One Search). Further reports were identified from citations of included papers and published literature (April 2020). Text-word searches along with MeSH terms or Subject Headings, were used to construct database searches (see Table 1). There were no date restrictions. All published reports were considered for inclusion if the abstract was in English. Searches of unpublished literature included relevant screening program reports. 

### 2.3. Inclusion and Exclusion Criteria

We included reports of programs with children with PCHL identified as a result of established UNHS programs with comparisons to children with PCHL identified as a result of targeted/risk screening, children with PCHL not identified through hearing screening programs, and children with PCHL identified as a result of distraction screening. Studies that included developmental outcomes of language, auditory/speech production/perception, literacy, social-emotional, cognitive, or quality of life were included. The eligibility criteria were selected to address the research questions with reference to Participants, Intervention, Comparators, Outcomes, Study designs, and Timings (PICOST) [14,15] as shown in Table 2. 

### 2.4. Article Selection and Data Extraction

Two independent evaluators extracted articles that met the inclusion criteria. First, articles were selected on the basis of titles and abstracts. Second, both independent evaluators read articles that appeared to have relevance to the systematic review questions and selected articles that met the criteria. Third, two independent evaluators compared selected articles and discussed articles that were not on both lists. A third reviewer (VM) evaluated all articles with a discrepancy between reviewers 1 and 2 for inclusion. After discussion, a final list of final articles was chosen. Figure 1 presents the PRISMA flow diagram of different phases through the systematic review. 

### 2.5. Synthesis of Included Studies

Most articles report very different developmental outcomes using a range of outcome measures. For this reason, a quantitative synthesis was not possible. A descriptive synthesis of the included articles was performed as outlined and described by Campbell et al. [15] and Popay et al. [16] to answer the specific questions. 

### 2.6. Quality Analysis and Level of Evidence

The Critical Appraisal Skills Programme (CASP) checklist was used to evaluate the quality of included studies [17]. CASP provides a specific checklist for different study designs (https://casp-uk.net/casp-tools-checklists/; accessed on 1 May 2021) and a relevant version of the checklist (cohort, case-control, and economic evaluation). The checklist contained questions on several sections that enable a structured approach to finding evidence, determining a possible sources of bias, and evaluating the internal and external validity of each study. 

Assessment of the level of evidence for each outcome was rated according to the Grading of Recommendations Assessment, Development and Evaluation (GRADE) protocol [18]. The GRADE level of evidence includes four levels: (a) high quality, (b) moderate quality, (c) low quality, or (d) very low quality. Higher scores are indicative of more confidence in the cumulative evidence. The appraisal of studies was performed independently by two researchers (CYI and VM) and any discrepancies were resolved by discussion with the third researcher (CH).

## 3. Results

### 3.1. Study Characteristics

Table 3 provides a summary of included studies as related to four key outcomes. Of the 30 included studies, 7 studies were focused on early identification and intervention, 11 studies were focused on developmental outcomes, 4 studies included cost-effectiveness, and 9 studies were focused on the adverse effects on parents of children with PCHL. 

#### 3.1.1. Question 1: Early identification and Intervention

##### Does UNHS Lower the Age of Identification and Increase the Number of Children Identified Early?

UNHS lowers the age of identification when compared to risk factor screen (RFS) and no screen (NS) groups. The research across the globe did not report their statistics consistently. Some research reported the proportion/percentage of the cohorts below and above a specific benchmark, whereas others have reported a mean/median for the different cohorts [19,22,23,25,38]. See Appendix A.

Cohort Comparisons: A prospective cohort study from the United Kingdom (UK) indicated that for children identified as a result of UNHS, the proportion of children for whom the age of identification was less than 9 months was 5 times greater than for those without UNHS [38]. The no-UNHS group had children identified through a distraction screen implemented in the UK after the children were 7 months of age. In a United States (US) study, Yoshinaga-Itano et al. [25] reported the results of the population of the state of Colorado *n* = 274. A matched sample of screen (*n* = 25) and NS (*n* = 25) groups had a median age of identification of 5 weeks for UNHS and 24 months for the NS group; 84% of the UNHS group were identified by 6 months while only 8% of the NS group were identified by 6 months of age. More recently, in an Australian quasi-randomized cohort study [22], a comparison was made between UNHS, RFS and NS screen and the results showed that the mean age of identification for UNHS was 9 months, (comparable to Kennedy et al. [38]) for RFS it was 18 months, and for NS it was 24 months. In a US cohort study, Sininger et al. [23] in California investigated a group of 64 children with hearing loss, 47 in the UNHS group and 17 in the NS group. The median age of identification for the UNHS group was 3 months (2.4 months for the Fail group and 22.5 months for the Pass group) and 27.8 months for the NS group. 

Single Program Descriptions: UNHS population statistics from the established program in the UK report that the median age of identification is significantly lower than the 1993–1996 UK birth cohort [38], as well as the no-screen cohort. Uus and Bamford [20] reported that the median age of identification for their UK cohort was 10 weeks. In another study, Weichbold and Wehlz-Mueller [39], a retrospective chart review of an Ear Nose and Throat department in Austria, reported a mean of 9.7 months for UNHS and 46 months for NS. The median for UNHS was 4 months as compared to a median 37.4 months for the NS group.

Population Description: Wood et al. [21] reported population statistics for the UK UNHS program from 2006–2013. The median age of identification was 49 days. 

Overall, the scientific evidence suggests that the UNHS lowers the age at which hearing loss is identified (see Appendix A). Regardless of the way that the data have been reported, all studies have found significantly lower ages of identification of congenital hearing loss. Most comparison studies found the age of identification with UNHS within the first few months of life possible, particularly with maturity of the UNHS system, while NS populations were identified at a mean/median age of 24 months or older. The systems across the world are quite different and the ages of identification vary dramatically by the date of publication and the newness of the UNHS program.

##### Does UNHS Lower the Age of Amplification Fit?

Comparisons between three Australian cohorts were made by Wake et al. [22], who found a lower age of amplification fit for UNHS 13.5 months when compared to 17.9 months in RFS group and also 24 months in NS group. Korver et al. [31] found that children in the Netherlands identified as a result of UNHS were fit, with an amplification at 15.7 months as compared to 29.2 months for children in the NS group. In the UK Wessex study, Kennedy et al. [19] reported a median age at amplification fit of 15 months for a 1993–1996 birth cohort. The UK has made significant progress towards lowering the age at amplification since 1993. Uus and Bamford [20] reported that a median age at amplification fit was 16 weeks. In a population study report for the UK for birth cohorts from 2006–2013, Wood et al. [21] reported a median age of amplification fit at 82 days. In a New York state study in the US, Dalzell, et al. [24] reported a median age of amplification fit at 7.5 months. Age of amplification for both UNHS and NS groups was within 3 months after the identification of hearing loss. Sininger et al. [23] reported a median age of amplification fit of 5.6 months for all UNHS and 4.9 months for the UNHS fails compared to a median of 29.1 months for the NS group. 

Overall, the studies that were reviewed included comparisons of cohorts of UNHS, targeted screened, distraction screened, and NS or opportunistic identification (see Appendix A). All studies regardless of the risk of bias or quality found that UNHS lowers the age at which children are fit with amplification. 

##### Does UNHS Lower the Age at Which Early Intervention Services Are Initiated?

Kennedy et al. [38], from a prospective cohort study in the UK, found that the odds of initiating education management had an 8 times greater chance of being initiated before 9 months with UNHS than without. Sininger et al. [23] reported a mean age of 10.58 months for early intervention service initiation, with a median of 8.9 months for UNHS and median 30.5 months for NS. In the US, Yoshinaga-Itano et al. [25] reported that 87% of all participants initiated early intervention within two months of the age of identification, which would be a mean of 3.1 months for UNHS as compared to a mean of 24 months for the NS groups. Dalzell et al. [24] reported a mean age of early intervention initiation at 3 months. Uus and Bamford [20] reported a mean of early intervention service initiation at 10 weeks (2.5 months) Wood et al. [21] reported a median age of referral to early intervention services of 50 days.

All studies showed that UNHS lowers the age at which early intervention services are initiated, whether they were cohort comparison studies, single cohorts after implementation of UNHS and population studies. In summary, as UNHS programs mature, the age of identification, age at amplification and age at initiation of early intervention services have dropped significantly. Refer to Appendix A for further details.

#### 3.1.2. Question 2: Developmental Outcomes

##### Does UNHS Improve Receptive and Expressive Language?

Kennedy et al. [38] reported higher adjusted mean z scores for language as compared with non-verbal ability for both confirmation by nine months of age and birth during UNHS, although z scores for expressive language were not significantly higher for children 5.4–11.7 years of age. Worsfold et al. [27] found that earlier confirmed hearing loss was associated with significantly more sentences and categories of high-pitched and morphological markers at a mean age of 7 years 7 months. The odds ratio for higher performance of early identified children was 3.03. No differences between early-identified and late-identified children were found for the number of categories of low-pitched morphological markers, poorer logical simplifications and sentences with multiple clauses. 

Wake et al. [22] compared the outcomes of UNHS with risk factor screen (RFS) at the test age 5–6 years of age for both and no screen (NS) (test age 7–8 years) for receptive, expressive language and receptive vocabulary. For children without intellectual disability, significantly higher language quotients for UNHS were found with 88.9 (UNHS), 83 (RFS), 81.8 (NS) for receptive language, 89.3 (UNHS), 80.7 (RFS), and 74.9 (NS) for expressive language, and 91.5 (UNHS), 83.8 (RFS) and 79.4 (NS) for receptive vocabulary. Significant differences were found between language levels and cognitive levels. 

Korver et al. [31] reported that when comparing outcomes of children at the test age 3–5 years participating in UNHS and those participating in distraction screen, children identified through UNHS did not show significantly better receptive and expressive language quotients or significantly more words produced. The age of identification of children through UNHS was far from the Early Hearing Detection and Intervention (EHDI) 1-3-6 (screen by 1 month, identify by 3 months and in early intervention by 6 months) guidelines. 

Sininger et al. [32] found that age-of-fit predicted better receptive language at a test age of 3–5 years. Each month lag accounted for a 0.17 months delay in receptive language and a 0.30 months lag in expressive language. 

Yoshinaga-Itano et al. [33] found that children between the age of 9 months to 5 years who were screened had a receptive language quotient mean of 82.9 and an expressive language quotient mean of 81.5 as compared to the NS group with a receptive language quotient mean of 62.1 and an expressive language quotient mean of 66.8. Children born before 1992 and UNHS had a total language quotient of 55.7. In addition, children who were born during a period of UNHS had a 95.5 word produced mean as compared to 14.5 word produced mean for the no screen group. Children born during periods of screen had a mean of 30 different words versus a mean of 7 different words for the NS group. Children identified as deaf or hard of hearing born in hospitals that screened versus hospitals that did not screen for hearing were 2.54 times more likely to have language quotients within the normal range 80 or greater; 82.4% of the children born in hospitals that screened had language quotients within the normal range compared to 68.4% of the children born in hospitals that did not screen for hearing. 

Overall, children identified through UNHS had better receptive/expressive language and receptive/expressive vocabulary as long as the children were early-identified before 9 months of age. Refer to Appendix A for further details

##### Does UNHS Improve Speech Perception and Speech Production?

Sininger et al. [32] found that age of fit predicted better speech discrimination from 3 to 5 years of age. Sininger et al. [32] also found that age of fit predicted better speech intelligibility. Both degree of hearing loss and age of fit predicted better word and sentence articulation from 3 to 5 years of age. Very few studies have demonstrated that UNHS improves speech perception and speech production and those that have are studies of children in the first five years of life. Overall, the literature indicates that the timeline for speech production and speech perception may be longer than the sensitive period for the development of language skills.

Speech scores did not differ significantly for either exposure to UNHS or early confirmation of hearing loss in the Kennedy et al. study for children tested at 5.4–11.7 years [19]. 

Yoshinaga-Itano et al. [33] reported that children aged 9 to 61 months of age who had been screened had significantly more consonants, consonant blends and better speech intelligibility than children who were not screened. See Appendix A for further details.

##### Does UNHS Improve Literacy?

The prospective Wessex study reported literacy results of children between 13 and 19 years of age [32]. The primary predictors of literacy for this group of children were age of identification, maternal level of education, cognitive level and degree of hearing loss. EID children were slightly more than one half a standard deviation (SD) (0.63 SD) below the mean of the normally hearing control. The LID children were almost 2 SDs below the mean of the normal hearing control children (1.74 SD). Participation in UNHS was no longer a predictor after age of identification was introduced as a predictor. In this cohort, UNHS was effective only if children were early-identified. The EID group maintained the same half SD gap found at the earlier age of 5.4–11.7 years [30]. EID in this study was before 9 months versus after 9 months of age. 

At the age of 6 to 10 years, McCann et al. [26] reported that UNHS participation predicted a higher aggregate reading (0.39 adjusted mean difference) and adaptive behavior communication scores, (0.51 adjusted mean difference) vs no UNHS. The EID by 9 months had significantly higher adjusted mean scores for aggregate reading (0.51), for basic reading (0.55), for reading comprehension (0.48) and adjusted mean on the communication scale (0.56) than the later-detection group. Benefits to reading and communication were partially mediated by better language ability (0.51) and communication (0.56) than the later-detection group. Benefits to reading and communication were partially mediated by better language ability. Overall, children identified through UNHS had significantly better reading comprehension from early childhood through late adolescence, as long as the children are EID at least before 9 months of age. Refer to Appendix A for further details.

##### Does UNHS Improve Social Development?

Korver et al. [31] found that children, assessed between 3 to 5 years of age, identified through UNHS, had social development quotients 8.8 quotient points higher than those identified through distraction testing. 

Stevenson et al. [28] found no significant differences in children with a mean age of 7 years and 11 months by age of confirmation before and after 9 months on daily living skills and socialization, though children with hearing loss had significantly lower functioning than children with typical hearing. Lower socialization scores for both children with hearing loss and those with typical hearing were related to language development. Stevenson et al. [29] examined the relationship between children 6 to 10 years at Time 1 and 13–20 years at Time 2 in children with hearing loss using spoken language and found significant relationships between language at Time 1 and emotional behavioral problems at Time 2. See Appendix A for further details.

##### Does UNHS Reduce Behavior Problems?

Stevenson et al. [28] for the Wessex study found no significant differences in behavior problems in children between 5 and 11 years of age, by age of identification less than 9 months of age versus greater than 9 months or participation in UNHS versus NS groups. Lower behavior problems were related to higher language levels. Level of behavior problems of the children with hearing loss was significantly higher than the children with normal hearing. Higher language levels were related to fewer behavior problems. Wake et al. [22] found no differences in behavior problems between children in the UNHS, RFS, and NS groups. Overall, these studies provide mixed evidence for a connection between UNHS and behavioral problems. See Appendix A for further details.

##### Does UNHS Improve Quality of Life?

Wake et al. [22] found no differences in quality of life between children in the UNHS, RFS and NS groups. In another study, Korver et al. [31] found that pediatric quality of life was 5.3 times higher for children identified through UNHS than distraction screen. Age of identification information was not provided. Age at amplification for UNHS was 15.7 months as compared to 29.2 months for distraction screen. No studies were found that compared the quality of life of children who were screened and met EHDI 1-3-6 guidelines. Those studies including children identified through UNHS had very late ages of identification and differed in the results. Wake et al. [22] showed no significant differences, while Korver et al. [31] showed better quality of life for children identified through UNHS than distraction screening. However, it is likely that the quality of life is associated with language, literacy, and vocational outcomes. See Appendix A for further details.

#### 3.1.3. Question 3: Cost Effectiveness

##### Is UNHS Cost Effective and Is There a Cost Benefit?

Mehl and Thomson [36] compared the cost of UNHS from 1992–1996. Costs of the Colorado UNHS included prevalence of hearing loss, false-positive rate, positive predictive value, and sensitivity of the screening. These UNHS costs were compared to the costs of other newborn screening for congenital diseases. The costs of early intervention and other educational costs were included. The cost of identifying each case of congenital hearing loss was approximately $9600 per child. If only half of the children who are deaf or hard of hearing realized some ultimate savings in school-based costs because of UNHS and early amplification, the UNHS program would recover all screening costs after only 10 years through subsequent savings in avoided intervention. Recovery of all initial costs (and subsequent cost savings) was independent of improved developmental outcomes. Mehl and Thomson [36] projected a 50 percent reduction in education costs. 

Keren et al. [37] worked from the assumption that education costs would be reduced by 10 percent if 50–70% had language in the typical range due to UNHS compared to 28–40% if children were later-identified and estimated a lifetime cost savings of $430,000 per individual.

Schroeder et al. [34] conducted a study in the UK of the actual economic costs of congenital bilateral PCHL for children between 7 and 9 years of age. Unit costs were applied to estimates of health, social, and broader resource use made by 120 children with PCHL and 65 children in a normally hearing comparison. The mean annual societal cost was £14,092 British pounds for children with PCHL compared with £4206 British pounds for children with normal hearing, a cost difference of £9885 British pounds. After adjusting for severity and other potential confounders in a linear regression model, mean societal costs among children with PCHL were reduced by £2553 British pounds for each unit increase in the z score for receptive language. Exposure to UNHS was associated with a smaller cost reduction of £2213 British pounds. The best estimate of annual cost saving of UNHS in middle childhood is 21% of the neonatal cost of UNHS per child with PCHL in the United Kingdom. If such an annual cost saving were generalizable across other years of the child’s school life, this would support an economic argument in favor of UNHS. 

Chorozoglou et al. [35] in a UK study of adolescents, aged 13–20 years (*n* = 110; 73 with PCHL and 37 with normal hearing), performed a follow-through of the Wessex study of 157,000 births in Southern England in which half were exposed to UNHS. The study found that the mean annual costs for PCHL were £15,914 British pounds as compared to the £5883 British pounds cost of children with normal hearing annually. The difference was an annual cost of £10,031 British pounds. Costs for the education of children with PCHL decreased by £1616 British pounds each year with an increase of one unit in receptive language z-score. 

Overall, studies in the US and the UK have demonstrated the cost-effectiveness of UNHS in terms of savings to society (see Appendix A). 

#### 3.1.4. Question 4: Adverse Effects

##### Does UNHS Cause Social and/or Emotional Harm (e.g., Worry, Stress, Anxiety) to Parents (Mothers)? 

Tueller and White [40] and Tueller [48] conducted a study of 192 mothers whose babies were screened in 11 hospitals in Utah. Of these 192 mothers, at Time 1 (immediately after UNHS), 83 had an initial pass, 34 had a Fail/Pass 2nd screen, 9 had Fail/Fail 2nd screen and 66 did not know the results of the screen. At Time 2 (within 6 weeks after the initial screen), 95 of 192 responded; 60/83 had an initial pass, 18/34 had Fail/Pass, 7/9 had Fail/Fail and 10/66 had an unknown result. Parental infant health concerns were assessed through a survey. Maternal anxiety and vulnerability were measured. Mothers worried as much about other aspects of the infant’s health as hearing concerns. Mothers of children who initially failed UNHS were slightly more worried, but the worry disappeared by Time 2. There was no significant difference in maternal anxiety between mothers whose infants passed and mothers whose infants failed UNHS. 

Watkins et al. [41] reported that 288 of 290 mothers enrolled in the study. Of these, 49 failed in both ears and 79 failed in one ear. In Stage 1, 288 surveyed within days of UNHS for screen attitudes, satisfaction and anxiety. In Stage 2, 57 were assessed at 6 weeks, and in Stage 3, 61 were assessed at 9 months. There was no significant difference between responses of the 288 surveyed immediately after UNHS and those assessed at 9-months. About 1% of participants expressed worry. Crockett et al. [42] sent invitations to 722 mothers. They had a 53% (*n* = 344) response rate. In Group 1, 103 passed at either Screen 1 or Screen 2. In Group 2, 81 passed at Screen 3. In Group 3, 105 referred on one ear at Screen 3 to audiological assessment. In Group 4, 55 referred at Screen 3 in both ears to audiological assessment. A measure of anxiety was sent 3 weeks after the screen. The return rate was 65% for Group 1 and 57% for Group 2. Mean anxiety levels were in the normal range. Anxiety and worry increased significantly as the number of tests increased. There was a significant interaction between the amount of worry and the mother’s belief that no clear response did not mean that the child had a hearing loss. Anxiety of mothers in Group 4 was related to this belief. 

Crockett et al. [43] compared UNHS and Health Visitor Distraction Testing (HVDT) responses and found that there were no significant differences in mother’s anxiety and worry. There was higher satisfaction from UNHS than from HVDT and higher positive attitudes after satisfactory screen result from UNHS than HVDT. Vohr et al. [44] invited 307 parents, at the first screen 80% and at the second screen 90% agreed to participate. 88–89% reported no maternal worry, whereas the remaining 9–10% reported being worried. Mothers at greater risk of being worried were socially disadvantaged and less aware of UNHS. Kolski et al. [45] compared maternal anxiety from a cohort of 3202 children screened at birth and 2588 screened at 2 months of age in France. Screening at birth coverage was significantly higher for the infant screened at birth (95.7%) compared to those screened at two months of age (64.2%). The false positive rate was 0.29% for the first strategy compared to 2.65% for screening at 2 months of age. One hundred and forty three mother-infant pairs participated in psychological assessment for maternal anxiety and quality of interaction. No significant differences were found for the NS group as compared to either of the two screening strategies. Khairi et al. [46] reported a study of 78 mothers whose infants had a positive finding for the first screen and 50 mothers prior to a second screen were assessed for maternal anxiety. Ten percent of the mothers were found to have moderate anxiety and 8% were found to have severe anxiety after the first screen which dropped to 4% before the 2nd screen. These findings were consistent with the study by Stuart et al. [47]. 

Overall, these studies investigating whether or not UNHS caused parental harm have found that very little stress, anxiety and worry have been identified whether utilizing customized non-standardized questionnaires or standardized instruments of maternal anxiety and stress (see Appendix A). The levels of parental stress and anxiety are comparable to those reported by mothers of healthy newborns without hearing loss and amount of concern about the hearing screening is of significantly less concern than concerns about other health matters.

### 3.2. Quality Analysis and Level of Evidence

#### 3.2.1. Quality Analysis of Included Studies

The quality assessment of individual studies was assessed using the CASP checklist (see Table 4). All cohort studies addressed a clearly focused issue; however, the quality of studies regarding other criteria in the checklist varied. In particular, in most studies it was unclear if they identified and accounted for confounding factors (e.g., socioeconomic factors, parental education, support at home, learning environment). In addition, outcome measurement bias was also noted in nearly half of the studies. Issues were also noted in terms of not following the subjects completely and for a long-enough duration. Moreover, nearly half of the studies’ results were not precise enough. The only case-control study found to have a minimum bias with the only limitation being treating the experimental group and control group equally. Overall, despite some issues identified, several high-quality studies exist with minimum bias. The economic evaluation studies were found to have several issues and a higher risk of bias. Four of the six studies did not fully account for all important and relevant resources required for cost-benefit analysis. Most of the studies fully adjusted the costs for different times, although such an adjustment was not performed adequately in three of the included studies. 

#### 3.2.2. Determination of Level of Cumulative Evidence

The quality of the evidence underpinning the recommendations was graded using the GRADE (see Table 5). The GRADE rating for “developmental outcomes,” “cost benefit,” and also “negative side effects” was very low. However, the GRADE rating for “early identification and intervention” was low. Almost all studies included in the review were observational in nature. Randomization and blinding were not possible given the nature of the intervention. The ratings were not downgraded for lack of blinding because the outcome was deemed to be less potentially influenced. In general, the studies for “developmental outcomes,” and “negative side effects” showed a limited bias and also showed moderate association with consistent results across studies, although some inconsistencies were noted for studies in the outcomes “early identification and intervention” and “societal cost-benefit”.

## 4. Discussion

The implementation of UNHS was started nearly three decades ago. However, only a few HICs provide UNHS universally. In most countries, especially LMIC, the newborn hearing screening (NHS) is not universal and is only available to some populations, primarily through private hospitals. This may be attributed to the lack of a strong policy to implement the UNHS as well as issues related to cost to healthcare. This comprehensive review provides an update on evidence related to outcomes of UNHS. 

Our review identified positive outcomes on all the four outcomes this review targeted. UNHS results in lower age of identification [22,25,38] and lower age of amplification, as well as lower age of initiation of early intervention services [22,25,38] when compared to targeted/risk screen, distraction screen and no screen/opportunistic identification. Developmental outcomes including speech perception/production, receptive/expressive language, literacy, social development, and quality of life were better in children identified early through UNHS are higher than those identified later [21,26,30,33,38]. In particular, children who are EID have significantly better literacy outcomes at both 5–11 years and 13–17 years of age than those that are LID [26,30]. Although the cost-effectiveness studies were limited to those from the UK and the US, they demonstrated the cost-effectiveness of UNHS in terms of savings to society [34,35,36,37]. Finally, the studies focusing on adverse effects show that the UNHS does not result in any significant harm to the parents of children who undergo UNHS [34,35,40,41,48]. Taken these studies together, there is long-standing evidence to demonstrate the positive outcomes of UNHS.

It is noteworthy that the outcome studies that were included were predominantly from very HIC countries. This is because the studies from LMIC did not meet the inclusion criteria primarily because they were descriptions of a single cohort. However, descriptive studies from lower resourced countries replicated many of the outcomes reported [49,50,51,52,53]. These studies have demonstrated that more recent UNHS programs that have been established around the world, especially in LMIC, are making progress towards lowering the age of identification, age of amplification, and age at intervention and meeting 1-3-6 goals, despite the challenging socio-economic situations.

### 4.1. Comparison of UNHS Outcomes of Other Newborn Screening Programs

Most newborn screening programs, e.g., genetic/metabolic, involve a blood sample with multiple tests run on the same sample. In these programs, intervention after identification is within the medical system and begins through the primary medical provider. Visual screening and intervention are completed within the medical system. UNHS begins with screening in the neonatal period followed by audiologic follow-through for diagnosis and fitting of amplification technology. However, the primary intervention is within the educational system, initially involves parent education and is followed by direct child intervention within the educational system through 18 years of age.

### 4.2. Limitations of Existing Evidence and the Review Process

Despite the large amount of literature on UNHS from all across the globe, there are several limitations to the existing literature that limit applicability. First, there is a limited number of controlled studies and the quality of the included studies varied widely as identified by the quality analysis of CASP ratings. There are no randomized controlled trials. This is because it is not ethical to deny the opportunity of early identification of hearing impairment for newborns which will result in life-long consequences. Moreover, the quality analysis identified that many studies did not consider the potential confounders and the long-term outcomes were reported by only a few studies. Additionally, pediatric hearing loss is a low-incidence disability and even with thousands of infants screened, only about two per thousand are identified with hearing loss. Creative ways are needed to perform more controlled studies (e.g., cluster randomization) to strengthen the evidence base in this area. Second, the choice of outcome measure as well as the reporting of outcomes varied widely across the studies. Expert consensus is needed to agree on key outcomes that should be measured and reported when performing UNHS. In addition, researchers should use a standard reporting format (e.g., EQUATOR network guidelines [54]) to ensure that as many details about the study findings are provided when publishing scientific findings in this area. Finally, most of the studies from LMIC countries did not have a control group and hence did not meet the eligibility criteria for this review. More work is needed to develop high quality studies from LMIC as the context could play an important role in the outcomes of UNHS.

The current review was much broader in focus compared to other reviews pertaining to UNHS [1,2]. This was necessary to answer the questions raised by the GDG committee of the WHO when making policy recommendations. However, this broad nature of review did not allow us to do an in-depth analysis of the included studies. Moreover, the heterogeneity of studies in terms of study design as well as outcomes reporting did not allow us to perform a quantitative synthesis of included studies.

## Figures and Tables

**Figure 1 jcm-10-02784-f001:**
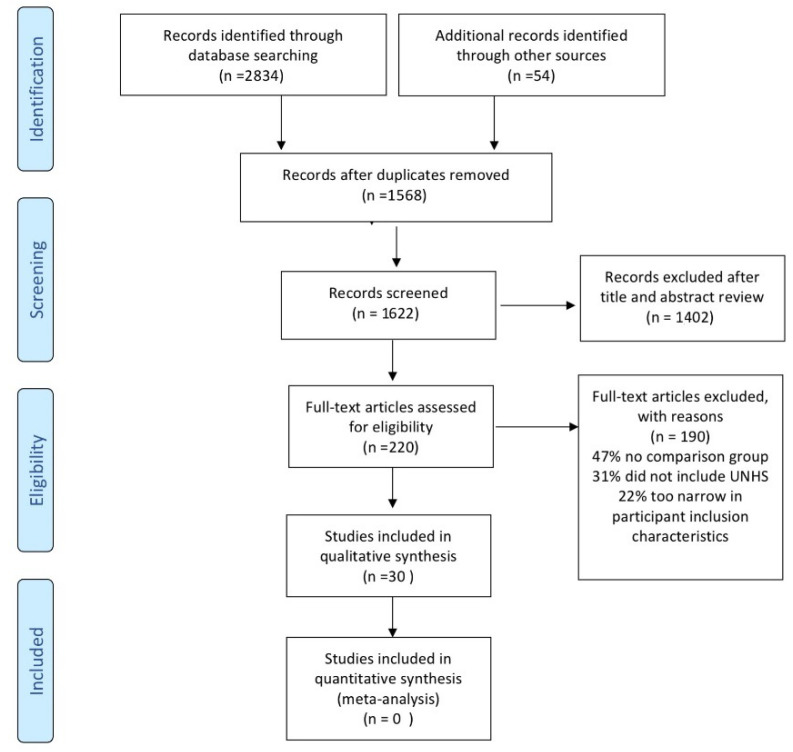
PRISMA systematic review flow diagram.

**Table 1 jcm-10-02784-t001:** Search strategy used.

**Concept 1**	**AND**	**Concept 2**	**AND**	**Concept 3**
(newborn hearing screening) OR (universal newborn hearing screening)		(hearing loss) OR (hearing impairment) OR (childhood hearing impairment) OR (permanent childhood hearing loss)		(outcome) OR (speech outcome) OR (language outcome) OR (literacy outcome) OR (maternal anxiety) OR (maternal stress)

**Table 2 jcm-10-02784-t002:** Inclusion and exclusion criteria using the Population, Intervention, Control, Outcomes, Study design and Timeframe (PICOST) criteria.

	Inclusion	Exclusion
Participants	Newborns (gestational age ≥37 weeks at birth without complications) and children within the first year of life undergoing Newborn Hearing Screening (NHS)/ Universal Newborn Hearing Screening (UNHS)	Only reporting screening of high-risk babies
Interventions	Universal newborn hearing screening	Target (or selective) screening with the UNHS with no comparison to UNHS
Comparators	No screening or selected/targeted screen	
Outcomes	▪ Outcomes of UNHS in terms of Early Hearing Detection and Intervention (EHDI): age of identification, age of amplification, age of intervention start. ▪ Developmental outcomes including receptive and expressive language, speech perception and speech production, literacy, social development, behavioral problems, quality of life. ▪ Cost benefit or cost-effectiveness or cost estimates for UNHS if available ▪ Adverse effects of UNHS on parents of children with hearing loss.	Not reporting the key outcomes listed
Study designs	▪ Studies regardless of design including those that were observational, including cohorts, case-controls or cross-sectional. ▪ Studies that include comparisons with controlled comparisons or that have done universal screening without controlled comparisons but have reported data for healthy and at-risk infants separately. ▪ Must have had NHS in place or being tested on the population. ▪ Must report the total number of children screened and the number of children with confirmed permanent bilateral hearing loss detected as a result of NHS or UNHS. ▪ Must have reported the total number of children undergoing NHS or UNHS screening.	▪ Reviews, comment pieces, letters or editorials ▪ Study reporting on a selective sample at risk of hearing loss without comparison with UNHS or healthy infants. ▪ Studies where the full spectrum of hearing loss was not represented. ▪ Studies where NHS/UNHS was not in place or being tested. ▪ Studies which did not specify the total number of children confirmed to have hearing loss at NHS/UNHS. ▪ Studies which did not specify the total number of children undergoing NHS/UNHS or with numerator bias.
Timings	No restrictions	No restrictions
Other criteria	English language articles	Non-English articles

**Table 3 jcm-10-02784-t003:** Types of outcomes of UNHS reported in the studies included (*n* = 30).

Study	Early Identification and Intervention			Developmental Outcomes								Cost Analysis	Adverse Effects
	Age of Identification	Age at Amplify	Intervention Start	Receptive Language	Expressive Language	Speech Perception	Speech Production	Literacy	Social dev	Behavioral Problems	Quality of Life		
**Question 1: Early identification and intervention (7 studies)**													
Kennedy et al. (2006) [19]	✓	✓	✓	✓	✓		✓						
Uus and Bamford (2006) [20]	✓	✓	✓										
Wood et al. (2015) [21]	✓	✓	✓										
Wake et al. (2016) [22]	✓	✓		✓	✓					✓	✓		
Sininger et al. (2009) [23]	✓	✓	✓										
Dalzell et al. (2000) [24]	✓	✓	✓										
Yoshinaga-Itano et al. (2001) [25]	✓	✓	✓	✓	✓		✓						
**Question 2: Developemental outcomes (11 studies, Kennedy et al., 2006 and Wake et al., 2016 listed earlier)**													
McCann et al. (2008) [26]				✓	✓			✓					
Worsfold et al. (2010) [27]				✓	✓								
Stevenson et al. (2010) [28]				✓	✓				✓	✓			
Stevenson et al. (2018) [29]				✓	✓			✓	✓	✓			
Pimperton et al. (2016) [30]				✓	✓			✓					
Korver et al. (2010) [31]				✓	✓				✓		✓		
Sininger et al. (2010) [32]				✓	✓	✓	✓						
Yoshinaga-Itano et al. (2000) [33]				✓	✓		✓						
Yoshinaga-Itano et al. (2020) [11]				✓	✓								
**Question 3: Cost effectiveness** **(4 studies)**													
Schroeder et al. (2006) [34]												✓	
Chorozoglou et al. (2018) [35]												✓	
Mehl and Thomson (2002) [36]												✓	
Keren et al. (2002) [37]												✓	
**Question 4: Adverse effects** **(10 studies)**													
Kennedy et al. (1998) [38]													✓
Weichbold and Welzl-Mueller (2001) [39]													✓
Tueller and White (2016) [40]													✓
Watkin et al. (1998) [41]													✓
Crockett et al. (2006) [42]													✓
Crockett et al. (2005) [43]													✓
Vohr et al. (2001) [44]													✓
Kolski et al. (2007) [45]													✓
Khairi et al. (2011) [46]													✓
Stuart et al. (2000) [47]													✓

**Table 4 jcm-10-02784-t004:** CASP ratings for included studies based on its design.

**a. CASP ratings for cohort studies (Y = Yes, C = Can’t tell, N = No, NA = Not applicable)**																									
**Article**	**1. Did the study address a clearly focused issue?**	**2. Was the cohort recruited in an acceptable way?**		**3. Was the exposure accurately measured to minimize bias?**		**4. Was the outcome accurately measured to minimize bias?**		**5. (a) Have the authors identified all important confounding factors?**		**5. (b) Have they taken account of the confounding factors in the design and/or analysis?**		**6. (a) Was the follow up of subjects complete enough?**	**6. (b) Was the follow up of subjects long enough?**		**7. What are the results of this study?**		**8. How precise are the results?**		**9. Do you believe the results?**		**10. Can the results be applied to the local population?**		**11. Do the results of this study fit with other available evidence?**		**12. What are the implications of this study for practice?**
Kennedy et al. (2006) [19]	Y	Y		Y		Y		C		C		Y	Y		Y		C		Y		Y		Y		Y
Uus and Bamford (2006) [20]	Y	Y		Y		Y		Y		Y		Y	Y		Y		Y		Y		Y		Y		Y
Wood et al. (2015) [21]	Y	Y		Y		Y		Y		Y		Y	Y		Y		Y		Y		Y		Y		Y
Wake et al. (2016) [22]	Y	Y		Y		Y		Y		Y		Y	C		Y		C		Y		Y		Y		Y
Sininger et al. (2010) [32]	Y	C		C		Y		C		C		C	Y		C		Y		Y		Y		Y		Y
Dalzell et al. (2000) [24]	Y	Y		Y		Y		Y		C		C	C		Y		Y		Y		Y		Y		Y
McCann et al. (2008) [26]	Y	Y		Y		Y		C		C		Y	Y		Y		Y		Y		Y		Y		Y
Worsfold et al. (2010) [27]	Y	Y		Y		Y		C		C		Y	Y		Y		C		Y		Y		Y		Y
Stevenson et al. (2010) [28]	Y	Y		Y		Y		C		C		Y	Y		C		C		Y		Y		Y		Y
Stevenson et al. (2018) [29]	Y	Y		Y		Y		C		C		Y	Y		Y		Y		Y		Y		Y		Y
Pimperton et al. (2016) [30]	Y	Y		Y		Y		C		C		Y	Y		Y		Y		Y		Y		Y		Y
Korver et al. (2010) [31]	Y	Y		C		C		C		C		Y	Y		C		C		Y		C		C		C
Sininger et al. (2009) [23]	Y	C		C		C		C		C		C	C		Y		C		Y		Y		Y		Y
Yoshinaga-Itano et al. (2020) [11]	Y	Y		Y		Y		C		C		Y	Y		Y		Y		Y		Y		Y		Y
Kennedy (1998) [38]	Y	Y		Y		Y		C		Y		Y	Y		Y		C		Y		Y		Y		Y
Weichbold and Welzl-Mueller (2001) [39]	Y	Y		Y		C		C		C		C	Y		C		C		Y		Y		Y		Y
Tueller and White (2016) [40] and Tueller (2006) [48]	Y	Y		Y		Y		Y		Y		Y	Y		Y		Y		Y		Y		Y		Y
Watkin et al. (1998) [41]	Y	Y		Y		Y		C		Y		Y	Y		Y		Y		Y		Y		Y		Y
Crockett et al. (2006) [42]	Y	Y		Y		C		Y		Y		C	C		Y		Y		Y		Y		Y		Y
Crockett et al. (2005) [43]	Y	C		Y		Y		C		C		Y	Y		Y		C		Y		Y		Y		Y
Vohr et al. (2001) [44]	Y	Y		C		C		C		C		C	C		Y		Y		Y		Y		Y		Y
Kolski et al. (2007) [45]	Y	Y		Y		Y		C		C		Y	C		Y		Y		Y		Y		Y		Y
Khairi et al. (2011) [46]	Y	Y		Y		C		C		C		C	C		Y		C		Y		Y		Y		Y
Stuart at al. (2000) [47]	Y	Y		Y		C		C		C		Y	C		Y		Y		Y		Y		Y		Y
**b. CASP ratings case-control studies (Y = Yes, C = Can’t tell, N = No, NA = Not applicable)**																									
**Article**	**1. Did the study address a clearly focused issue?**		**2. Did the authors use an appropriate method to answer their question?**		**3. Were the cases recruited in an acceptable way?**		**4. Were the controls selected in an acceptable way?**		**5. Was the exposure accurately measured to minimize bias?**		**6. (a) Aside from the experimental intervention, were the groups treated equally?**			**6. (b) Have the authors taken account of the potential confounding factors in the design and/or in their analysis?**		**7. How large was the treatment effect?**		**8. How precise was the estimate of the treatment effect?**		**9. Do you believe the results?**		**10. Can the results be applied to the local population?**		**11. Do the results of this study fit with other available evidence?**	
Yoshinaga-Itano et al. (2001) [25]	Y		Y		Y		Y		Y		C			Y		Y		Y		Y		Y		Y	
Yoshinaga-Itano et al. (2000) [33]	Y		Y		C		C		Y		Y			C		Y		Y		Y		Y		Y	
**c. CASP ratings economic evaluation studies (Y = Yes, C = Can’t tell, N = No, NA = Not applicable)**																									
**Article**	**1. Was a well-defined question posed?**		**2. Was a comprehensive description of the competing alternatives given?**		**3. Does the paper provide evidence that the programme would be effective? (i.e., would the programme do more good than harm?)**		**4. Were the effects of the intervention identified, measured and valued appropriately?**		**5. Were all important and relevant resources required, and health outcome costs for each alternative identified, measured in appropriate units and valued credibly?**		**6. Were costs and consequences adjusted for different times at which they occurred (discounting)?**			**7. What were the results of the evaluation?** **their analysis?**		**8. Was an incremental analysis of the consequences and cost of alternatives performed?**		**9. Was an adequate sensitivity analysis performed?**		**10. Is the programme likely to be equally effective in your context or setting?**		**11. Are the costs translatable to your setting?**		**12. Is it worth doing in your setting?**	
Schroeder et al. (2006) [34]	Y		Y		Y		Y		Y		C			Y		C		NA		Y		Y		Y	
Chorozoglou et al. (2018) [35]	Y		Y		Y		Y		Y		C			Y		C		NA		Y		Y		Y	
Mehl and Thomson (2002) [36]	Y		Y		Y		C		C		C			Y		C		N		Y		Y		Y	
Keren et al. (2002) [37]	Y		Y		Y		Y		C		N			Y		Y		NA		Y		C		Y	

**Table 5 jcm-10-02784-t005:** GRADE level of evidence for UNHS outcomes.

Certainty Assessment							№ of Patients		Effect		Certainty	Importance
№ of Studies	Study Design	Risk of Bias	Inconsistency	Indirectness	Imprecision	Other Considerations	UNHS	No Screen and/or Targeted Screen	Relative (95% CI)	Absolute (95% CI)		
**Early identification and intervention (follow up: median 3 years)**												
7	observational studies	not serious	serious	not serious	not serious	strong association all outcomes indicate lower age ID, age amp, and age Rx	5,882,275	596,874	not estimable	NA	⨁⨁◯◯ LOW	CRITICAL
**Developmental outcomes (follow up: median 3 years)**												
11	observational studies	serious	serious	not serious	serious	strong association all plausible residual confounding would reduce the demonstrated effect	725,037	1,556,737	not estimable	NA	⨁◯◯◯ VERY LOW	CRITICAL
**Cost-benefit (follow up: median 5 years)**												
4	observational studies	serious	serious	not serious	not serious	none	68,714	88,019	not estimable	NA	⨁◯◯◯ VERY LOW	CRITICAL
**Negative side effects (follow up: median 3 months)**												
10	observational studies	serious	not serious	not serious	not serious	none	717,826	88,019	not estimable	NA	⨁◯◯◯ VERY LOW	IMPORTANT

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
