# Peer review of "Outcomes of Universal Newborn Screening Programs: Systematic Review"

_jcm, 2021, doi:10.3390/jcm10132784_

Round 1

Reviewer 1 Report

Thank you for the opportunity to review the “Outcomes of universal newborn hearing screening systematic review”.

General feedback: The paper had a well-developed aim and justification for undertaking the work. Good study design was provided, with logical decisions made on criteria to include and exclude. The results were presented in a logical and easy to interpret format. Discussion summarised the paper well. My main concern with this paper is that the wording errors, issues with sentence structure, and formatting inconsistencies throughout the paper made it difficult to read. I have noted a few, however, I feel a thorough review of readability is required.  

Feedback on each section is below. 

INTRODUCTION

Line 39-41: wording error – please review sentence.

Line 41-44: This sentence doesn’t make sense, please review.

Line 50-51: wording error – please review

Line 63: pleaser insert the word “for” before UNHS

Justification for the study was presented. The aim was clearly stated.

METHOD

Methodology was sound.

Line 77-79: Please reword definition to “All children with bilateral hearing loss of 20 dB in the better ear…….

Line 88: “reports” not “report”

Line 94: please include “through” between the words “identified hearing screening”

Line 92-93: states that reports were included if children were identified through UNHS with comparisons to children with PCHL identified as a result of targeted/risk screening, yet, exclusion criteria stated reports with high risk screening and target screening were excluded. How come?

Table 2: EHDI not in abbreviations list

Line 118: please replace the comma with a full stop.

Line 126: please review for readability

RESULTS

For all questions, the results were analysed appropriately into study categories with clear and concise conclusions drawn. The quality of the studies and levels of evidence measures used were of a high standard.  

Line 163: grammatical error

Line 181” remove the word “the”

Line 225: Acronym used before first stating in full

Line 228-229: acronyms already used

Line 358: PCI?

DISCUSSION

Discussion provided a brief summary of the findings from the review.

Line 526: compared, not compares

Author Response

REVIEWER 1 review:

Thank you for the opportunity to review the “Outcomes of universal newborn hearing screening systematic review”.

General feedback: The paper had a well-developed aim and justification for undertaking the work. Good study design was provided, with logical decisions made on criteria to include and exclude. The results were presented in a logical and easy to interpret format. Discussion summarised the paper well. My main concern with this paper is that the wording errors, issues with sentence structure, and formatting inconsistencies throughout the paper made it difficult to read. I have noted a few, however, I feel a thorough review of readability is required.  

Feedback on each section is below. 

INTRODUCTION

AUTHORS’ RESPONSES TO THE REVIEW

Line 39-41: wording error – please review sentence.  Corrected

Line 41-44: This sentence doesn’t make sense, please review. Corrected

CORRECTION:  Prior to establishing Universal Newborn Hearing Screening (UNHS) programs, the average language, literacy, social-emotional and speech development of children with permanent childhood hearing loss (PCHL) was significantly lower than their peers with normal hearing. Eighteen year old children with hearing loss in the United States who were in the 12th grade between 1974-2003, had average literacy proficiency, between 3rd and 4th grade levels, more than two standard deviations below the developmental functioning of their hearing peers [3].

Line 50-51: wording error – please review Corrected

UNHS programs began to be implemented in the early 1990s, and by the end of the 1990s, there was evidence that these programs resulted in earlier identification of hearing loss, earlier amplification, earlier enrollment into early intervention services and significantly improved developmental outcomes in early childhood [5].

Line 63: pleaser insert the word “for” before UNHS Corrected

Justification for the study was presented. The aim was clearly stated.

METHOD

Methodology was sound.

Line 77-79: Please reword definition to “All children with bilateral hearing loss of 20 dB in the better ear…….Corrected

All children with bilateral hearing loss of 20 dB or greater in the better ear were included in studies reviewed within this systematic review. 

Line 88: “reports” not “report” Corrected

Line 94: please include “through” between the words “identified hearing screening” Corrected

Line 92-93: states that reports were included if children were identified through UNHS with comparisons to children with PCHL identified as a result of targeted/risk screening, yet, exclusion criteria stated reports with high risk screening and target screening were excluded. How come? High risk and targeted screening were only excluded if they had no comparison group with children with UNHS,  Table 2 was corrected to add “with no comparison to UNHS”

Table 2: EHDI:  Early Hearing Detection & Intervention and UNHS/NHS:  Universal Newborn Hearing screening/Newborn Hearing Screening       - was added to the bottom of the Table

Line 118: please replace the comma with a full stop. Corrected

Line 126: please review for readability  Corrected

Higher scores represent more confidence in the cumulative evidence.

RESULTS

For all questions, the results were analysed appropriately into study categories with clear and concise conclusions drawn. The quality of the studies and levels of evidence measures used were of a high standard.  

Line 163: grammatical error Corrected

Line 181” remove the word “the”  Corrected

Line 225: Acronym used before first stating in full Corrected

Line 228-229: acronyms already used Corrected

Line 358: PCI?should have been PCHL  Corrected

DISCUSSION

Discussion provided a brief summary of the findings from the review.

Line 526: compared, not compares  Corrected

Other grammatical errors were corrected in the text.

Submission Date

15 May 2021

Date of this review

28 May 2021 03:55:26

Reviewer 2 Report

  1. line 41-44

Did the research cover children who had only prelingual hearing loss without any additional burdens? Did the children have hearing aids?

The research conducted in Poland on children with prelingual deafness, without additional disabilities, who have received a cochlear implant, show the dependence of the achieved success in learning from the moment of cochlear system implantation. Implantation until the second year of life results in better development [Zgoda M., Lorens A., Obrycka A., Skarżyński H.: Academic achievement of polish children with cochlear implants at the end of their primary education. Journal of Hearing Science. 2019; 9(1): 25-31.]

  1. line 78-79

Which classification was applied to assess the hearing loss ?

According to ASHA, the hearing norm is up to 15 dB HL inclusive, according to BIAP up to 20, and according to WHO up to 25 dB HL

I think it is worth to add a citation on the classification used for the analyses, or cite the classifications used in the reviewed articles

  1. Regarding the questions asked in the manuscript: “Whether UNHS enhances…” suggests that the program forces use of/implantation of hearing prostheses. From what I know, however, I can say it is only an early diagnostics aiming to detect the hearing loss and enabling the early intervention of the hearing prosthetist. The parent is not forced to do so and many factors are considered.

I believe the world improve should be replaced with contribute.

  1. line 218-265

(as above) From this paragraph one can conclude that hearing screening contributes to language improvement, and it is not true, as the language skills development is connected with early intervention such as provision of hearing aids. Unfortunately you cannot conclude it from the text, there is too big mental shortcut. The hearing screening itself is not a treatment. Much depends on a properly selected hearing prosthesis, rehabilitation, and parents. If screening is performed and the parents fail to take further steps, the child will not develop proper speech.

Remark to further questions – it should be clearly ststed that UNHS is one thing and the assistance of hearing aids is another thing. Without a hearing aid, there would be no improvement in any of these spheres of life

Only the following sentence in the Discussion indicates the place of UNHS in the entire diagnostic process, and it should be clear from the very beginning.

I am quoting the authors: UNHS results in lower age of identification [19-21], lower age of amplification as well as lower age of initiation of early intervention services [19-21] when compared to targeted/risk screen, distraction screen and no screen/opportunistic identification.

  1. Bibliography remark:

The bibliography should be corrected according to the indications of the hjournal (below indications from the JCM website). There is no appropriate style, moreover, the names of the magazines are not abbreviated. Please standardize the spelling of DOI.

Author Response

REVIEWER 2

  1. line 41-44

The reported studies conducted between 1974 and 2003 with the Stanford Achievement Test included children with prelingual hearing loss who were judged by their teachers to be capable of taking the national reading examination. 

Did the research cover children who had only prelingual hearing loss without any additional burdens? Did the children have hearing aids?

The research conducted in Poland on children with prelingual deafness, without additional disabilities, who have received a cochlear implant, show the dependence of the achieved success in learning from the moment of cochlear system implantation. Implantation until the second year of life results in better development [Zgoda M., Lorens A., Obrycka A., Skarżyński H.: Academic achievement of polish children with cochlear implants at the end of their primary education. Journal of Hearing Science. 2019; 9(1): 25-31.]

  1. line 78-79

Which classification was applied to assess the hearing loss ?

Because UNHS screening technology was not capable of screening for hearing loss to 15 dB HL, the definition of hearing loss reported by all the UNHS studies was 20 dB HL or greater. At the beginning of UNHS in the late 1990s, the purpose of UNHS was defined as identifying hearing loss of 40 dB HL or greater. The definitions of hearing loss are based on behavioral thresholds.  For follow-up after UNHS, behavioral thresholds are not obtainable in newborns and physiological thresholds were used to identify hearing loss.  There is no convention for the determination of hearing loss utilizing physiological thresholds.  Each study used a different definition of hearing loss.

According to ASHA, the hearing norm is up to 15 dB HL inclusive, according to BIAP up to 20, and according to WHO up to 25 dB HL

I think it is worth to add a citation on the classification used for the analyses, or cite the classifications used in the reviewed articles

  1. Regarding the questions asked in the manuscript: “Whether UNHS enhances…” suggests that the program forces use of/implantation of hearing prostheses. From what I know, however, I can say it is only an early diagnostics aiming to detect the hearing loss and enabling the early intervention of the hearing prosthetist. The parent is not forced to do so and many factors are considered.

I believe the world improve should be replaced with contribute.

For this commissioned systematic review, WHO defined intervention as UNHS and the approved PROSPERO protocol was a systematic review to identify variables that improved outcome. 

We are, however, in complete agreement with the reviewer that without the amplification technology intervention and the early intervention services, following UNHS, there would be no “improved” outcomes.  Fortunately, the UNHS studies defined their EHDI systems as including these variables and that contributed significantly to the outcomes reported. 

  1. line 218-265

(as above) From this paragraph one can conclude that hearing screening contributes to language improvement, and it is not true, as the language skills development is connected with early intervention such as provision of hearing aids. Unfortunately you cannot conclude it from the text, there is too big mental shortcut. The hearing screening itself is not a treatment. Much depends on a properly selected hearing prosthesis, rehabilitation, and parents. If screening is performed and the parents fail to take further steps, the child will not develop proper speech.

Remark to further questions – it should be clearly ststed that UNHS is one thing and the assistance of hearing aids is another thing. Without a hearing aid, there would be no improvement in any of these spheres of life

The authors completely agree with Reviewer 2.  However, the WHO systematic review that was commissioned defined the screening as the intervention.  While we indicated that screening with the intervention of amplification and early intervention services was insufficient, we were unable to convince the WHO post-natal guidelines physicians that screening was not the intervention.  However, fortunately the members of the WHO clinical guidelines working group indicated understanding that screening without amplification and early intervention services would not be successful. 

Only the following sentence in the Discussion indicates the place of UNHS in the entire diagnostic process, and it should be clear from the very beginning.

I am quoting the authors: UNHS results in lower age of identification [19-21], lower age of amplification as well as lower age of initiation of early intervention services [19-21] when compared to targeted/risk screen, distraction screen and no screen/opportunistic identification.

We added this introductory statement to make it clear that UNHS/EHDI is a system of delivery and not just screening.

“UNHS/EHDI programs are systems that begin with the neonatal screening but must lead to earlier identification of the hearing loss, earlier access to amplification, and early intervention services if the implementation of UNHS/EHDI is to lead to improved developmental outcomes for children who are deaf or hard of hearing.”

  1. Bibliography remark:

The bibliography should be corrected according to the indications of the hjournal (below indications from the JCM website). There is no appropriate style, moreover, the names of the magazines are not abbreviated. Please standardize the spelling of DOI

The bibliography was corrected to the JCM reference style.  The spelling of “doi” has been standardized.  The names of journals and magazines have been abbreviated. 

Submission Date

15 May 2021

Date of this review

01 Jun 2021 14:20:00